mental disorders; intentional self-harm; epidemiology; umbrella review; South Asia

**Corresponding author:**
Aishwarya L. Vidyasagaran;
Email: aishwarya.vidyasagaran@york.ac.uk

# Prevalence of mental disorders in South Asia: A systematic review of reviews

Aishwarya L. Vidyasagaran[1] , David McDaid[2] , Mehreen R. Faisal[1],
Muhammad Nasir[3], Krishna P. Muliyala[4], Sreekanth Thekkumkara[5], Judy Wright[6],
Rumana Huque[7], Saumit Benkalkar[8] and Najma Siddiqi[1]

[1]Department of Health Sciences, University of York, Heslington, UK; [2]Care Policy and Evaluation Centre, Department of Health Policy, London School of Economics and Political Science, London, UK; [3]Department of Economics, Institute of Business Administration (IBA), Karachi, Pakistan; [4]Department of Psychiatry, National Institute of Mental Health & Neurosciences (NIMHANS), Bengaluru, India; [5]School of Health Sciences, University of Dundee, Dundee, UK; [6]Leeds Institute of Health Sciences, University of Leeds, Leeds, UK; [7]ARK Foundation, Dhaka, Bangladesh and [8]King's College London, London, UK

## Abstract

Mental disorders are increasing in South Asia (SA), but their epidemiological burden is under-researched. We carried out a systematic umbrella review to estimate the prevalence of mental disorders and intentional self-harm in the region. Multiple databases were searched and systematic reviews reporting the prevalence of at least one mental disorder from countries in SA were included. Review data were narratively synthesised; primary studies of common mental disorders (CMDs) among adults were identified from a selected subset of reviews and pooled. We included 124 reviews. The majority ($n = 65$) reported on mood disorders, followed by anxiety disorders ($n = 45$). High prevalence of mental disorders and intentional self-harm was found in general adult and vulnerable populations. Two reviews met our pre-defined criteria for identifying primary studies of CMDs. Meta-analysis of 25 primary studies showed a pooled prevalence of 16.0% (95% CI = 11.0–22.0%, $I^2 = 99.9\%$) for depression, 12.0% (5.0–21.0%, $I^2 = 99.9\%$) for anxiety, and 14.0% (10.0–19.0, $I^2 = 99.9\%$) for both among the general adult population; pooled estimates varied by country and assessment tool used. Overall, reviews suggest high prevalence for mental disorders in SA, but evidence is limited on conditions other than CMDs.

## Impact statement

Our umbrella review provides the most comprehensive estimates for the prevalence of mental disorders and intentional self-harm in South Asia (SA) and highlights that large proportions of the population in the region (both general-adult and specific vulnerable groups) are affected by these adverse health conditions. Evidence is critically lacking beyond common mental disorders on several conditions including schizophrenia and psychotic disorders, behavioural syndromes, personality disorders and intellectual disabilities. Although limited by heterogeneity and methodological quality of included studies, our review findings show an urgent need for countries in SA to formulate and implement clinical and policy measures for the prevention and early treatment of mental disorders and intentional self-harm. The pooled prevalence estimated for depression and anxiety in the general-adult population could serve as a reference for policy-makers to take necessary action for curbing the growing burden of mental disorders in SA.



## Introduction

Mental disorders are recognised to be increasing globally, and contribute to a growing health, social and economic burden (World Health Organization, 2021). From 1990 to 2019, they have gone from the 13th to the 7th leading cause of disease burden in the world, with the number of disability-adjusted life-years (DALYs) due to mental disorders increasing from 80.8 million to 125.3 million; they also remain the second largest contributor to years lived with disability (GBD Mental Disorders Collaborators, 2022). Intentional self-harm accounts for a further 34.1 million DALYs, with their burden being greatest in low- and middle-income countries (Knipe et al., 2022). In South Asia (SA) (Afghanistan, Bangladesh, Bhutan, India, Maldives, Nepal, Pakistan and Sri Lanka) (The World Bank, 2019), rapid demographic and lifestyle changes are said to be associated with an exponential rise in mental and substance-use disorders, which health systems and services are unable to adequately meet (World Health Organization, 2016b; Ambekar et al., 2019). This has resulted in a considerable mental health treatment gap, with more than 75% of people affected in

many countries not having access to the treatment they need (World Health Organization, 2016b; Gautham et al., 2020). Further, mental disorders have not been a policy priority among countries in the region, and their epidemiological and psychosocial burdens have been under-researched (Shidhaye et al., 2015). To address these issues and to improve the knowledge base for better planning and decision-making, an overall evaluation of the prevalence of mental disorders and intentional self-harm among countries in SA is needed.

Hossain et al. (2020) published an umbrella review, stating its advantages over a review of primary studies for understanding the population-level burden of mental disorders within the SA region. However, their inclusion criteria were limited to reviews solely conducted in SA (i.e., they excluded broader reviews, even if those reviews included some South Asian studies). We considered that expanding the scope of our umbrella review to identify all systematically conducted reviews, so long as they included evidence from at least one country in SA (whilst limiting our synthesis to South Asian studies), would provide a more complete picture of the prevalence of mental disorders and intentional self-harm in the region. In addition, a meta-analysis to provide an updated pooled estimate for the prevalence of mental disorders in the general adult population in SA would complement the overview provided by the umbrella review.

## Methods

The review was registered with PROSPERO (CRD42021282957) (McDaid et al., 2021). We followed the Joanna Briggs Institute (JBI) method for conducting the review (Aromataris et al., 2015) and the PRISMA guidelines for reporting (Page et al., 2021; Supplementary Appendix 1).

### Search strategy

We searched multiple electronic databases and research repositories, covering published and grey literature, on 29 September 2021 (Supplementary Appendix 2). Our searches included index terms, synonyms, and alternative phrases to cover mental disorders, South Asian countries, prevalence or epidemiology, and review types. We used the search strategies for 'prevalence' and 'South Asia' from Uphoff et al. (2019), and for 'mental disorders' from Mishu et al. (2021), adapting them to include all ICD-10 categories of mental disorders and intentional self-harm (World Health Organization, 2016a; Supplementary Appendix 3). Searches were developed by an information specialist (JW) and peer-reviewed by a second, using the PRESS checklist (McGowan et al., 2016). There were no limits for language or publication date. We also screened reference lists and forward citations of included studies. In addition, PROSPERO records were checked for any relevant ongoing or completed reviews. Retrieved records were de-duplicated in EndNote semi-automatically, using specified guidance (AUHE Information Specialists, 2016) and uploaded to COVIDENCE (www.covidence.org) for further evaluation.

### Inclusion criteria and study selection

We included systematic reviews (with or without meta-analyses) that searched two or more databases, and provided keyword and/or search strategies, as per the quality criteria of the AMSTAR2 checklist (Shea et al., 2017). Reviews reporting the prevalence or incidence of mental disorders in one or more countries in the World Bank-defined SA region were eligible. This included reviews that had data from countries beyond SA, but where we could extract the SA data on their own. All populations and settings were eligible, except studies of international military forces based in SA. Reviews on any mental, behavioural, and neurodevelopmental disorders (ICD-10, F-codes), or on suicide and intentional self-harm (ICD-10, X60-X84 codes) were eligible (Supplementary Appendix 4). Two authors independently evaluated all records at title and abstract and full-text screening stages. Discrepancies in screening were addressed through discussion with a third author.

### Data extraction and synthesis

A pre-piloted data extraction tool was uploaded to COVIDENCE. Two authors independently extracted data and performed quality appraisals for 10% of included reviews, with good agreement; discrepancies were identified and resolved through consensus. All remaining extractions were performed by a single author and checked by a second. Extraction items included objective and type of review, year of publication, name and timeframe of databases, originating countries of primary studies, sample size and characteristics, as well as reported prevalence or incidence of mental disorders. We used the AMSTAR2 tool for evaluating the methodological quality of included reviews (Shea et al., 2017).

Narrative synthesis was conducted according to the type of review (with or without meta-analysis) and mental disorders (ICD-10 categories), using tables and figures. For the reviews that went beyond SA, we only considered the pooled/range of estimates from the subgroup of studies that were relevant to the SA region. Next, we focused on the reviews with meta-analyses to summarise results for pooled prevalence of mental disorders in SA. Finally, to estimate prevalence for the common mental disorders (CMDs), depression and anxiety, we obtained data from primary studies in the included reviews. We limited this step to reviews with a pre-registered protocol (as a quality indicator), and those reporting on CMDs in the general adult population, given these conditions, which comprise the great majority of mental disorders, were the focus of the bulk of included reviews. Additional primary studies reporting CMD prevalence in SA were identified through forward citation screening of included reviews, to capture more recent studies.

Data extraction from primary studies was again performed by a single author and checked by a second on the following items: country, state or province of the study population, sample characteristics and sample size, and prevalence or incidence for each mental disorder. For quality assessment, we used the JBI Critical Appraisal Checklist for prevalence studies (Munn et al., 2014), but did not exclude ones at high risk of bias from further analysis. We created a 'summary of findings' table for primary studies and carried out meta-analyses using Stata (2007), Version 17.0 to produce a pooled estimate of prevalence for depression and anxiety among the general population in SA. Heterogeneity was assessed using $I^2$ statistics, and subgroup analyses based on country and outcome ascertainment tools were conducted to explore the sources. Evidence of publication bias was assessed using funnel plots and Egger's test.

## Results

Our searches yielded 1,048 records, with 770 remaining after deduplication (Figure 1). Following title and abstract screening,

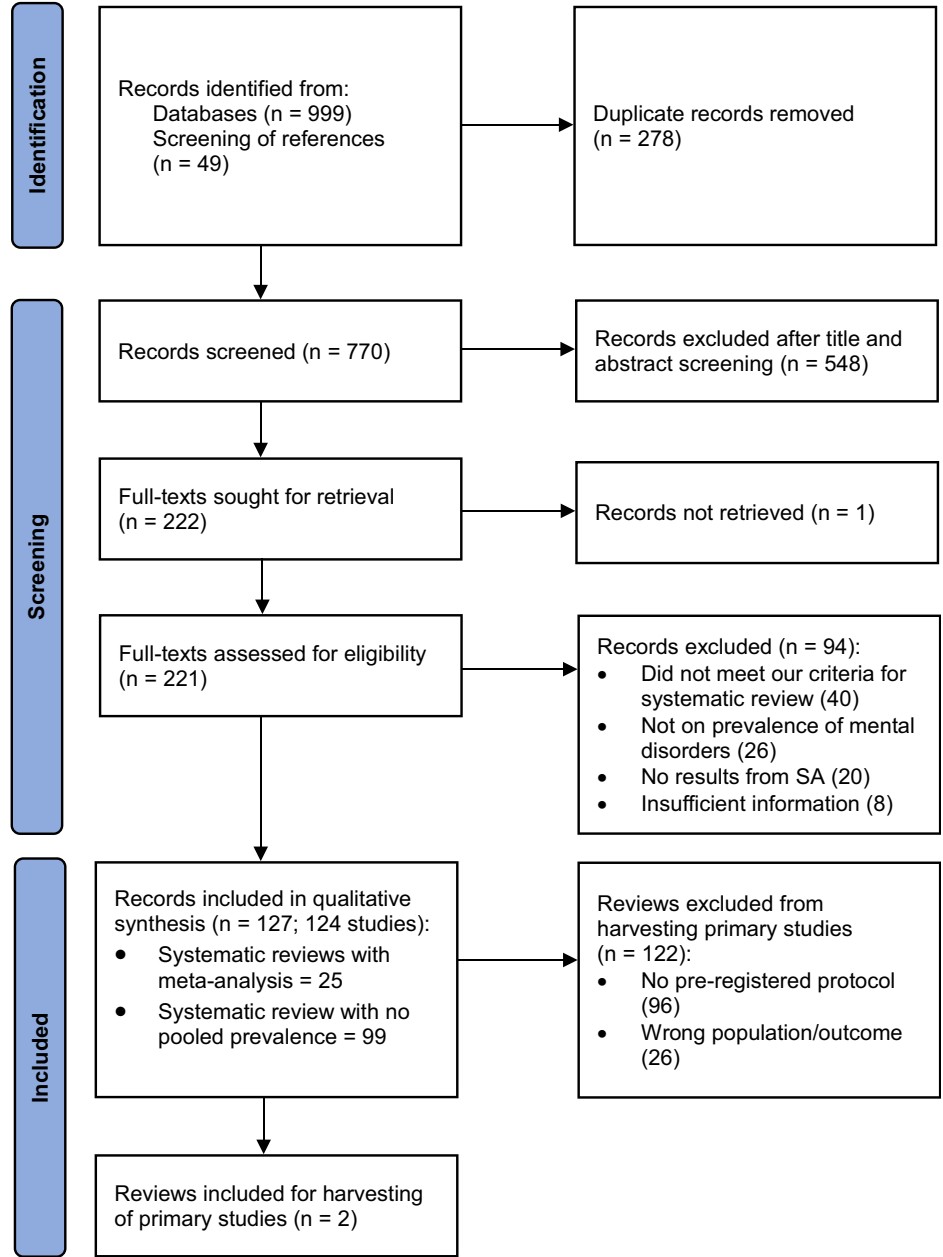

**Figure 1.** PRISMA flow chart for included reviews.

548 records were excluded, and all but one of the remaining 222 papers were obtained. Full-text screening resulted in the exclusion of 94 records (see Figure 1 and Supplementary Appendix 5 for details). 124 reviews (127 records) met our eligibility criteria and were included in the narrative synthesis. Three reports covering one review were merged (Barua et al., 2010; Barua et al., 2011a, 2011b); for another review, we merged and extracted data from both the original and updated reports (Oram et al., 2012; Ottisova et al., 2016).

For the meta-analyses of primary studies, we found only two reviews with pre-registered protocols, which reported on the prevalence of CMDs in the general adult population (Naveed et al., 2020; Zuberi et al., 2021). These provided 22 primary studies. Three additional studies were identified through forward citation screening of included reviews, resulting in 25 distinct primary studies for our meta-analyses (14 depression-only, three anxiety-only, and eight both) (see Supplementary Appendix 6 for flow chart of primary studies).

### Characteristics of reviews included in the review of reviews

Table 1 provides the summary characteristics of all included reviews. Twenty-five reviews had conducted meta-analyses providing pooled estimates for mental disorders in SA. A further 99 reviews did not provide pooled estimates, either because no meta-analysis was conducted ($n = 61$), no pooled values limited to SA countries were presented ($n = 37$), or pooled prevalence was not estimated ($n = 1$).

The earliest review was published in 2004 (Mirza and Jenkins, 2004), with the majority ($n = 116$) published after 2010. The number

**Table 1.** Summary characteristics of included reviews

| Reference | South Asian countries represented in the review (number of primary studies), sample size | Population | Mental disorder(s) and Prevalence/other measure of burden reported | AMSTAR-2 grading |
|---|---|---|---|---|
| Systematic reviews with meta-analysis (*n* = 25); reporting pooled prevalence (95% CI) unless otherwise specified | | | | |
| Barua et al. (2010, 2011a, 2011b) | India (6); *n* = 2,499 | Older people | Depression 21.9% (11.6–31.1) | Critically low |
| Steel et al. (2014) | Afghanistan (1), Bangladesh (1), India (7), Pakistan (3); *n* = 17,524 | General adult population | CMD 19.8% (10.3–34.7) | Critically low |
| Cho et al. (2016) | India (2), Pakistan (1), Sri Lanka (1); *n* = 327 | People with suicidal behaviour | Any mental disorder among fatal suicide 90.4% (71.8–97.2) | Low |
| Ranjan and Asthana (2017) | Afghanistan (1), Bangladesh (5), Bhutan (1), India (20), Nepal (3), Pakistan (3), Sri Lanka (1); *n* = 158,555 | General adult population | Any mental disorder (rate per 1000) 122.0 (8.0–252.0) | Critically low |
| Upadhyay et al. (2017) | India (38); *n* = 20,043 | Postpartum women | Depression 22.0% (19.0–25.0) | Low |
| Hussain et al. (2018) | India (37); *n* = 10,270 | People with type 2 diabetes | Depression 38.0% (31.0–45.0) | Moderate |
| Chauhan et al. (2019) | India (4); *n* = 130,599 | Children | ASD 0.1% (0.0–0.2) | Low |
| Hendrickson et al. (2019) | India (22); *n* = 5,122 | Adults with AUD | Mood disorder 18.0% (5.6–45.1), Anxiety disorder 2.4% (0.9–5.8) | Low |
| Mahendran et al. (2019) | Bangladesh (3), India (12), Maldives (1), Nepal (1), Pakistan (14), Sri Lanka (2); *n* = 13,087 | Pregnant women | Depression 24.3% (19.0–30.5) | Moderate |
| Pilania et al. (2019) | India (51); *n* = 22,005 | Older people | Depression 34.4% (29.3–39.7) | High |
| Prabhu et al. (2019) | Bangladesh (1), India (12), Maldives (1), Nepal (5), Pakistan (9); *n* = 15,345 | Postnatal women | Depression 26.0% (21.0–30.0) | Critically low |
| Uphoff et al. (2019) | Bangladesh (5), India (60), Pakistan (30), Multi-country (1); *n* = NR | Adults with NCD | Depression 41.0% (37.0–44.0), Anxiety 29.0% (22.0–36.0) | Moderate |
| Ganesan et al. (2020) | India (10); *n* = 6,513 | Children and adolescents | Suicide attempt past-year 0.6% (0.0–1.8), lifetime 17.1% (5.0–35.4) | Low |
| Khan et al. (2020) | Pakistan (26); *n* = 7,652 | University students | Depression 42.7% (34.8–50.9) | Moderate |
| Naveed et al. (2020)[a] | Bangladesh (8), India (81), Nepal (20), Pakistan (33), Sri Lanka (12), Multi-country (6); *n* = NR for all studies, range: 250 to 863,657 | Adults – general, students, older people | CMD 28.4% (13.9–49.3), Alcohol abuse 12.9% (8.8–18.6), Opiates misuse 0.8% (0.2–2.5), Drug abuse 2.5% (0.1–32.1), Depression 26.4% (23.6–29.4), Bipolar 0.6% (0.3–1.0), Anxiety 25.8% (19.4–33.5), Panic disorder 1.3% (0.5–3.4), Phobias 1.8% (0.4–7.1), OCD 1.6% (0.4–5.5), PTSD 17.2% (11.0–25.9), Suicidal behaviour 6.4% (3.1–12.4) | Moderate |
| Abraham et al. (2021) | Pakistan (15); *n* = 2,890 | HCWs | Depression 31.7% (18.7–48.3) | High |
| Assariparambil et al. (2021) | Bangladesh (7), India (89), Nepal (12), Pakistan (6), Sri Lanka (6); *n* = 65,060 | Older people | Depression 42.0% (38.0–46.0) | Low |
| Atif et al. (2021) | Pakistan (43); *n* = 17,544 | Perinatal women | Depression antenatal 37.0% (30.0–44.0), postnatal 30.0% (25.0–36.0) | Low |
| Choudhary et al. (2021) | India (20); *n* = 86,312 | Older people | Dementia 2.0% (2.0–3.0) | Critically low |
| Hossain et al. (2021) | Bangladesh (7), India (19), Nepal (3), Pakistan (5), Sri Lanka (1); *n* = 41,402 | General population and HCWs, COVID-19 | Depression 34.1% (28.9–39.4), Anxiety 41.3% (34.7–48.1) | Low |
| Hosseinnejad et al. (2021) | Pakistan (6); *n* = 3,403 | General population, after earthquakes | PTSD 49.2% (39.4–59.0) | Low |
| Kalra et al. (2021) | India (27); *n* = 7,880 | Antenatal women | CMD 21.9% (17.5–26.3) | Moderate |
| Patra et al. (2021) | India (15); *n* = 1,617 | Stroke survivors | Depression 55.0% (43.0–65.0) | Low |
| Yadav et al. (2021) | India (10); *n* = 2,362 | Peri-menopausal women | Depression 42.5% (28.7–57.5) | High |

(*Continued*)

**Table 1.** (*Continued*)

| Reference | South Asian countries represented in the review (number of primary studies), sample size | Population | Mental disorder(s) and Prevalence/other measure of burden reported | AMSTAR-2 grading |
|---|---|---|---|---|
| Zuberi et al. (2021)[a] | Afghanistan (2), Pakistan (5); $n$ = 19,314 | General adult population | Afghanistan: SUD 0.0% (0.0–1.0), Depressive disorder 33.0% (7.0–75.0), Bipolar 0.0% (0.0–3.0), Anxiety disorder 25.0% (6.0–62.0), OCD 1.0% (0.0–5.0), Panic disorder 0.4% (0.1–2.0) PTSD 35.0% (4.0–87.0); Pakistan: SUD 32.0% (6.0–78.0), Depressive disorder 10.0% (4.0–25.0), Anxiety disorder 4.0% (0.0–27.0) | Moderate |
| Systematic reviews with no pooled estimates ($n$ = 99), reporting prevalence/prevalence range unless otherwise specified | | | | |
| Mirza and Jenkins (2004) | Pakistan (20); $n$ = 9,170 for 17 relevant studies | General adult population | CMD 33.6% | Critically low |
| Mills et al. (2005) | India (5); $n$ = 410 | Tibetan refugee population | MDD 11.5–57.0%, Anxiety 25.0–77.0%, PTSD 11.0–23.0% | Low |
| Collins et al. (2006) | India (7), Nepal (1); $n$ = 281 | People with HIV/AIDS | Any mental disorder PWA 75.0% and HIVP 47.6%, Alcohol dependency 44.4%, Psychosis 5.0%, Depression 3.0–47.0%, Anxiety 25.0–36.0%, Adjustment disorder 27.8%, Suicidal intention/attempt 14.0% | Low |
| Lopes et al. (2007) | India (2); $n$ = 2,603 | Older people | Dementia 1.3–3.1% | Critically low |
| Mills et al. (2008) | Nepal (6); $n$ = 4,712 | Bhutanese refugee population | Depression 2.0%, Anxiety 4.0%, Phobia 18.5%, Dissociative disorder 8.0%, PTSD 25.0%, Somatoform pain 31.0% | Critically low |
| Klainin and Arthur (2009) | India (3), Nepal (1), Pakistan (3); $n$ = 2,072 | Postpartum women | Depression 4.9–56.0% | Critically low |
| Math and Srinivasaraju (2010) | India (16); $n$ = 72,202 | General adult population | Any disorder (rate per 1000) 9.5–102.0 | Critically low |
| Das and Leibowitz (2011) | India (NR); $n$ = NR | People with HIV/AIDS | Depression 33.0–70.0%, Anxiety 25.0–36.0%, Adjustment disorder 27.8%, Persistent suicidal intent/attempt 14.0% | Critically low |
| Maulik et al. (2011) | Bangladesh (2), India (1), Pakistan (2), Multi-country (1); $n$ = 6,09,731 | General population | Intellectual disability (rate per 1000) 0.9–156.0 | Low |
| Fisher et al. (2012) | Bangladesh (4), India (4), Nepal (2), Pakistan (4); $n$ = 5,126 | Perinatal women | CMD antenatal 11.5–33.0% and postnatal 9.0–59.4% | Critically low |
| Hawton et al. (2013) | India (5); $n$ = 649 | Persons with self-harm | Depressive disorder 53.0–89.0% | Critically low |
| Jones and Coast (2013) | Bangladesh (1), India (3), Nepal (1), Pakistan (3); $n$ = 2,479 | Postpartum women | Depression 4.9–35.6% | Low |
| Nadkarni et al. (2013) | India (31); $n$ = NR for all studies; range: 100 to 7,554 | Over 50 years | AUD 1.1–70.0% | Critically low |
| Newman (2013) | Bangladesh (61); $n$ = 12,021 for 16 relevant studies | General, 15 years and older | Depression 6.6–97.0% | Critically low |
| Rajapakse et al. (2013) | Sri Lanka (23); $n$ = 74,482 | General or clinical population | Intentional self-poisoning (rate per 100,000) 21.5–224.0 | Low |
| Udina et al. (2013) | India (11), Sri Lanka (1); $n$ = 799 | Adult males | Dhat syndrome ~7% of patients seen at sexual health clinics; Depression 24.0–66.0%, Anxiety 13.0–37.0% | Critically low |
| Beckwith et al. (2014) | India (1), Pakistan (1); $n$ = 16,318 | Mental health outpatient | Personality disorder 1.0–60.0% | Critically low |
| de Bernier et al. (2014) | India (4); $n$ = 5,616 | General or clinical, adults | Personality disorders 1.3–52.0% | Critically low |
| Fuhr et al. (2014) | India (7), Nepal (1), Pakistan (1), Sri Lanka (2); $n$ = NR | Perinatal women | Injury 1.1–17.9%, Suicide 1.0–10.7% | Moderate |
| Hossain et al. (2014) | Bangladesh (32); $n$ = 25,767 | General or clinical population | Any mental disorders 6.5–31.4% | Critically low |
| Jordans et al. (2014) | India (45), Bangladesh (26), Sri Lanka (18), Nepal (12), Pakistan (11), Afghanistan (1), Multi-country (1); $n$ = NR | General or clinical population | Suicide (incidence per 100,000) 0.43–331.0 | Moderate |

**Table 1.**  (*Continued*)

| Reference | South Asian countries represented in the review (number of primary studies), sample size | Population | Mental disorder(s) and Prevalence/other measure of burden reported | AMSTAR-2 grading |
|---|---|---|---|---|
| Medlow et al. (2014) | India (1); *n* = 150 | Homeless adolescents | Depression 8.0% | Critically low |
| Mendenhall et al. (2014) | Bangladesh (3), India (8), Pakistan (3); *n* = NR | People with type 2 diabetes | Depression 14.7–84.0% | Critically low |
| Pearson et al. (2014) | Sri Lanka (149); *n* = NR | General or clinical population | Suicide (rate per 100,000, as figure) ~25.0 | Low |
| Rane and Nadkarni (2014) | India (36); *n* = NR | General or clinical population | Suicide (rate per 100,000) 82.0–95.0 | Critically low |
| Aggarwal and Berk (2015) | India (27); *n* = 36,838 | Adolescents | Depression 0.5–60.0%, GAD 13.0%, Social anxiety disorder 12.8%, PTSD 29.0%, Behavioural problems 1.8–24.7%, Suicidal behaviour 3.9–25.4% | Critically low |
| Malakouti et al. (2015) | Pakistan (2); *n* = 2,663 | General population | Suicide (rate per 100,000) 0.6–1.1 | Critically low |
| Norhayati et al. (2015) | Bangladesh (3), India (2), Nepal (2), Pakistan (2); *n* = 2,545 | Postpartum women | Depression 3.1–59.4% | Critically low |
| Evagorou et al. (2016) | India (2), Nepal (1), Pakistan (1); *n* = 826 | Postpartum women | Depression 4.9–63.0% | Critically low |
| McKenzie et al. (2016) | India (1); *n* = 70,302 | General population | Intellectual disability (as figure) 1.0–1.2% | Low |
| Ottisova et al. (2016) | Nepal (1); *n* = 164 | Victims of human trafficking | Depression 86.0%, Anxiety 90.2%, PTSD 13.4% | Moderate |
| Sahu et al. (2016) | India (12); *n* = 547 | Amputees | Depression 10.4–63.0%, GAD 3.4–10.0%, PTSD 3.3–56.3% | Critically low |
| Tanzil Jamali (2016) | Pakistan (8); *n* = NR | Children and adolescents | Learning disability 24.8%, Emotional or behavioural disorders 34.0% | Critically low |
| Aggarwal et al. (2017) | India (2); *n* = 1,675 | 12–25 year olds | Non-suicidal self-harm 31.2%, Suicidal behaviour 6.1%, Suicide attempt 3.5% | Critically low |
| Ahmed et al. (2017) | Bangladesh (1), India (12), Sri Lanka (3); *n* = 3,024 | People with suicidal behaviour | Depression among those who died by suicide 6.9–37.1%, attempted 20.7–59.7% | Low |
| Dennis et al. (2017) | Bangladesh (2); *n* = 1,394 | Perinatal women | Anxiety 38.3%, Trait anxiety 29.4% | Low |
| Hossain et al. (2017) | Bangladesh (3), India (2), Sri Lanka (1); *n* = 41,620 | Children and adolescents | ASD 0.1–1.1% | Critically low |
| Kuppili et al. (2017) | India (73); *n* = 16,073 | Children and adolescents | ADHD 4.7–29.2% | Critically low |
| Naskar et al. (2017) | India (41); *n* = 34,119 | People with type 1&2 diabetes | Depression 2.0–84.0% | Critically low |
| Salmanian et al. (2017) | Afghanistan (1); *n* = 1,011 | Children and adolescents | Conduct disorder 4.8% | Low |
| Singh and Balhara (2017) | India (52); *n* = NR | People with cannabis use and psychiatric disorders | High frequency of psychiatric symptoms with SUDs, preponderance of cannabis-associated psychotic & affective disorders | Critically low |
| Woody et al. (2017) | India (3), Nepal (1), Sri Lanka (1); *n* = NR | Perinatal women | Depression NR for countries in SA | Critically low |
| Yatan Pal Singh et al. (2017) | India (13), Nepal (3); *n* = 51,008 | General or clinical population | AUD 3.9–100%, Depression 2.7–94.3% | Critically low |
| Halim et al. (2018) | Bangladesh (3), India (2), Nepal (2), Pakistan (3); *n* = 4,546 | Perinatal women | Depression antenatal 18–33% and postnatal 5–36%, Antenatal anxiety 29%, CMD 16–42%, Suicide attempts 2–5% | Critically low |
| Hunt et al. (2018) | India (1), Sri Lanka (1); *n* = 412 | People with psychosis | AUD 3.0–11.0%, CUD 20.0% | Low |
| Jha et al. (2018) | Bangladesh (1), India (4), Pakistan (2), Sri Lanka (1); *n* = 3,323 | Antenatal women | Depression 1.9–65.0%, Anxiety 26.0–49.0% | Low |
| Morina et al. (2018) | Nepal (2), Sri Lanka (2); *n* = 2,950 | Refugee and IDP | Depression 22.0–80.0%, MDD 5.0–8.0%, Anxiety 33.0–81.0%, PTSD 3.0–53.0% | Critically low |

**Table 1.** (*Continued*)

| Reference | South Asian countries represented in the review (number of primary studies), sample size | Population | Mental disorder(s) and Prevalence/other measure of burden reported | AMSTAR-2 grading |
|---|---|---|---|---|
| Morina et al. (2018) | Afghanistan (1), India (1), Sri Lanka (1); $n = 18,886$ | Civilian war survivors in area of conflict | Major depression 26.0–37.0%, PTSD 28.0–34.0% | Moderate |
| Shekhani et al. (2018) | Pakistan (110); $n = $ NR for 2 relevant studies | General or clinical population | Suicide (incidence per 100,000) 0.43–2.86 | Critically low |
| Shorey et al. (2018) | India (1), Nepal (1), Pakistan (3); $n = 1,329$ | Postpartum women | Depression 5.0–62.0% | Moderate |
| Thapa et al. (2018) | Nepal (32); $n = 4,152$ | Older people | Depressive disorders 4.4–53.2%, Anxiety 21.7–32.3% | Critically low |
| Arafat (2019) | Bangladesh (18); $n = 14,942$ for 3 relevant studies | General population | Suicide (rate per 100,000) 30.0–128.8 | Critically low |
| Bhagavathula et al. (2019) | India (17), Pakistan (4); $n = 4,441$ | People with hair dye poisoning | Suicide intent 75.0–99.9% | Critically low |
| Gilmoor et al. (2019) | India (56); $n = 38,932$ | General or clinical population | PTSD 0.1–89.0% | Critically low |
| Knipe et al. (2019) | Bangladesh (2), India (28), Nepal (2), Pakistan (2), Sri Lanka (5); $n = 9,888$ | People with suicidal behaviour | Any mental disorder among fatal suicide 48.0–96.0% and attempted 0.0–96.0% | High |
| Mytton et al. (2019) | Nepal (186); $n = $ NR | General or clinical population | Self-harm NR for countries in SA | Critically low |
| Somrongthong et al. (2019) | India (8); $n = 326$ for 1 relevant study | Female sex workers 10–19 years | Suicidal attempts 41.0% | Critically low |
| Tay et al. (2019) | Bangladesh (1); $n = 148$ | Rohingya refugee population | Depression 89.0%, PTSD 36.0% | Critically low |
| Vaidyanathan et al. (2019) | India (39); $n = 6,663$ | General or clinical population | Probable ED 4.0–45.4%, ED 1.25% | Critically low |
| Abate et al. (2020)[a] | India (3), Pakistan (3); $n = 1,312$ | People undergoing surgery with anaesthesia | Preoperative anxiety 24.0–88.0% | Moderate |
| Akhtar et al. (2020) | India (1), Nepal (2), Pakistan (1), Sri Lanka (1); $n = 7,495$ | University students | Depression 9.3–53.1% | Moderate |
| Banerjee et al. (2020) | Bangladesh (1), India (11), Pakistan (1); $n = 7,936$ | General or clinical, COVID-19 | Depression 10.5–34.9%, Anxiety 38.2–39.5% | Critically low |
| Blackmore et al. (2020) | Nepal (1); $n = 574$ | Adult refugees | Depression 1.9%, Anxiety 4.7%, PTSD 26.8% | Moderate |
| Devarapalli et al. (2020) | India (32); $n = $ NR for all studies; range: 103 to 114,068 | Tribal population | Depression 8.3%, Anxiety 6.4%, Adjustment disorder 9.0%, Somatoform pain 14.0%, PTSD 9.6%, Alcohol abuse 36.2%, Binge eating 6.4%, Bulimia nervosa 1.4%, Self-harm 11.2%, Suicide 14.2% | Critically low |
| Dua and Grover (2020) | India (33); $n = 13,227$ | Clinical population (liaison psychiatry settings) | Delirium 2.8–43.4%, Dementia 0.9–3.8%, SUD 1.8–28.9%, Organic psychosis 0.6–25.5%, Psychotic illness 3.2–33.3%, Depression 1.5–24.4%, Bipolar 2.3–10.4%, Anxiety 1.1–13.1%, Adjustment 0.4–16.0%, Dissociation 0.9–8.3%, Psychosomatic 0.8–7.7%, Psychosexual 0.7%, Personality disorder 0.6–5.3%, Mental retardation 0.6–7.0%, Conduct disorder 0.8%, ADHD 0.4–0.8%, Self-harm 2.7–33.9% | Critically low |
| Fekadu Dadi et al. (2020) | NR; $n = $ NR | Antenatal women | Depression NR for countries in SA | Moderate |
| Gilan et al. (2020) | India (1); $n = 662$ | General or clinical, COVID-19 | Hypochondriac fear 37.8% | Low |
| Hunt et al. (2020) | Sri Lanka (1); $n = 109$ | People with MDD | AUD 21.1%, CUD 1.8% | Critically low |

**Table 1.** (*Continued*)

| Reference | South Asian countries represented in the review (number of primary studies), sample size | Population | Mental disorder(s) and Prevalence/other measure of burden reported | AMSTAR-2 grading |
|---|---|---|---|---|
| Janse Van Rensburg et al. (2020) | India (11), Pakistan (3), Sri Lanka (1), Multi-country (1); *n* = NR | People with tuberculosis | AUD 4.0–58.0%, Depression 8.5–84.0%, Anxiety 2.0–47.2%, | Critically low |
| Kalra et al. (2020) | Bangladesh (8), India (24), Nepal (3), Pakistan (7); *n* = 12,650 | Adults with type 2 diabetes | Depression 11.6–67.5% | Critically low |
| Karimi et al. (2020) | India (1); *n* = 133 | People with migraine | Anxiety 16.54% | Low |
| Khunsa Junaid (2020) | India (2); *n* = 8,484 | HCWs, COVID-19 | Depression 34.8% | Low |
| Lasheras et al. (2020) | India (1); *n* = 250 | Medical students, COVID-19 | Anxiety 17.2% | Low |
| Liu et al. (2020) | Sri Lanka (1); *n* = 335 | People with history of deliberate self-harm | Non-fatal repetition of self-harm (incidence) 3.0% | High |
| Qiu et al. (2020) | Bangladesh (1), India (1), Nepal (2); *n* = 43,401 | Children and adolescents | ASD 0.1–0.3% | Low |
| Rahele et al. (2020) | Pakistan (1), Sri Lanka (1); *n* = 1,786 | Perinatal women, COVID-19 | CMD 14.3%, Depression 19.5%, Anxiety 17.5% | Critically low |
| Winsper et al. (2020) | Bangladesh (1); *n* = 766 | 12–18-year-olds | Personality disorder 0.5% | High |
| Yan et al. (2020) | Sri Lanka (1); *n* = 257 | Perinatal women, COVID-19 | Depression 28.0%, Anxiety 26.0% | Moderate |
| Al Falasi et al. (2021) | India (1); *n* = 426 | HCWs, COVID-19 | PTSD 7.3% | Low |
| Al Mamun et al. (2021) | Bangladesh (9); *n* = 18,201 | General or clinical, COVID-19 | Suicidal behaviour 6.1% | Critically low |
| Amiri and Behnezhad (2021) | Bangladesh (1), India (2), Nepal (1); *n* = 1,034 | Postpartum women | Suicide attempt 4.0–18.0% | Critically low |
| David Franciole de Oliveira et al. (2021) | India (1); *n* = 100 | Teachers, COVID-19 | CMD NR for countries in SA | Low |
| Dong et al. (2021) | India (1); *n* = 50 | People with COVID-19 | Depression 24.0%, Anxiety 32.0% | Moderate |
| Dutta et al. (2021) | India (4), Nepal (1), Pakistan (2); *n* = 1,869 | HCWs, COVID-19 | Depression 28.2–72.3%, Anxiety 34.0–85.7% | Moderate |
| Fellmeth et al. (2021) | India (7); *n* = 1,003 | Perinatal women | Depression 12.5–18.0% | High |
| Ghazanfarpour et al. (2021) | Pakistan (1), Sri Lanka (1); *n* = 1,786 | Pregnant women, COVID-19 | Depression 19.5%, Anxiety 14.3–17.5% | Critically low |
| Hosen et al. (2021) | Bangladesh (24); *n* = 49,806 | General or clinical, COVID-19 | Depression 12.1–82.4%, Anxiety 10.6–81.8%, PTSD/Stress 11.1–85.6% | Critically low |
| Jephtha and Jagadeesan (2021) | India (1); *n* = 15,981 | HCWs, COVID-19 | CMD NR for countries in SA | Critically low |
| Kar et al. (2021)[a] | Bangladesh (1) India (5) Pakistan (2); *n* = NR | Adult males | Dhat syndrome 64.6% | Critically low |
| Liu et al. (2021) | India (1); *n* = 662 | General, COVID-19 | Anxiety 58.5% | Moderate |
| Mahadevan et al. (2021) | Bangladesh (1), India (11), Sri Lanka (1); *n* = 2,013 | Stroke survivors | Depression 13.8–100.0%, Anxiety 80.9% | Moderate |
| Mahmud et al. (2021) | Bangladesh (2), India (7), Nepal (1), Pakistan (3); *n* = 5,422 | HCWs, COVID-19 | Depression 37.5–53.6%, Anxiety 41.9–62.2% | Moderate |
| Mamun (2021) | Bangladesh (7); *n* = 21,534 | Students, COVID-19 | Depression 46.9–82.4%, Anxiety 26.6–96.8% | Critically low |
| Mohammadi et al. (2021) | India (3), Sri Lanka (1); *n* = 3,757 | Children and adolescents | Conduct disorder 1.0–7.0% | High |
| Necho et al. (2021) | India (1); *n* = 662 | General adult, COVID-19 | CMD NR for countries in SA | Low |
| Panda et al. (2021) | Bangladesh (1), India (1); *n* = 505 | Children, adolescents and caregivers, COVID-19 | Anxiety, depression and/or sleep disturbance 57.0–68.0% | Low |

(*Continued*)

**Table 1.** (*Continued*)

| Reference | South Asian countries represented in the review (number of primary studies), sample size | Population | Mental disorder(s) and Prevalence/other measure of burden reported | AMSTAR-2 grading |
|---|---|---|---|---|
| Santabárbara et al. (2021)[a] | Bangladesh (2), India (2); *n* = 4,092 | General, COVID-19 | Anxiety 28.0–43.0% | Moderate |
| Vanderkruik et al. (2021) | Bangladesh (4); *n* = NR | Adolescents | Depression during pregnancy 7.0–14.0% and postpartum 10.4–36.2% | Moderate |
| Wang et al. (2021) | Bangladesh (2); *n* = NR | College students, COVID-19 | Depression 47.0–82.0%, Anxiety 33.0–84.0% | Critically low |

Abbreviations: ADHD, attention deficit hyperactivity disorder; ASD, autism spectrum disorder; AUD, alcohol use disorder; CI, confidence interval; CMD, common mental disorder; CUD, cannabis use disorder; ED, eating disorder; GAD, generalised anxiety disorder; HCW, health care worker; HIVP, HIV positive; IDP, internally displaced population; MDD, major depressive disorder; NCD, non-communicable disease; NR, not reported; OCD, obsessive compulsive disorder; PTSD, post-traumatic stress disorder; PWA, people with AIDS; SA, South Asia; SUD, substance use disorder.[a]Reporting discrepancy noted.

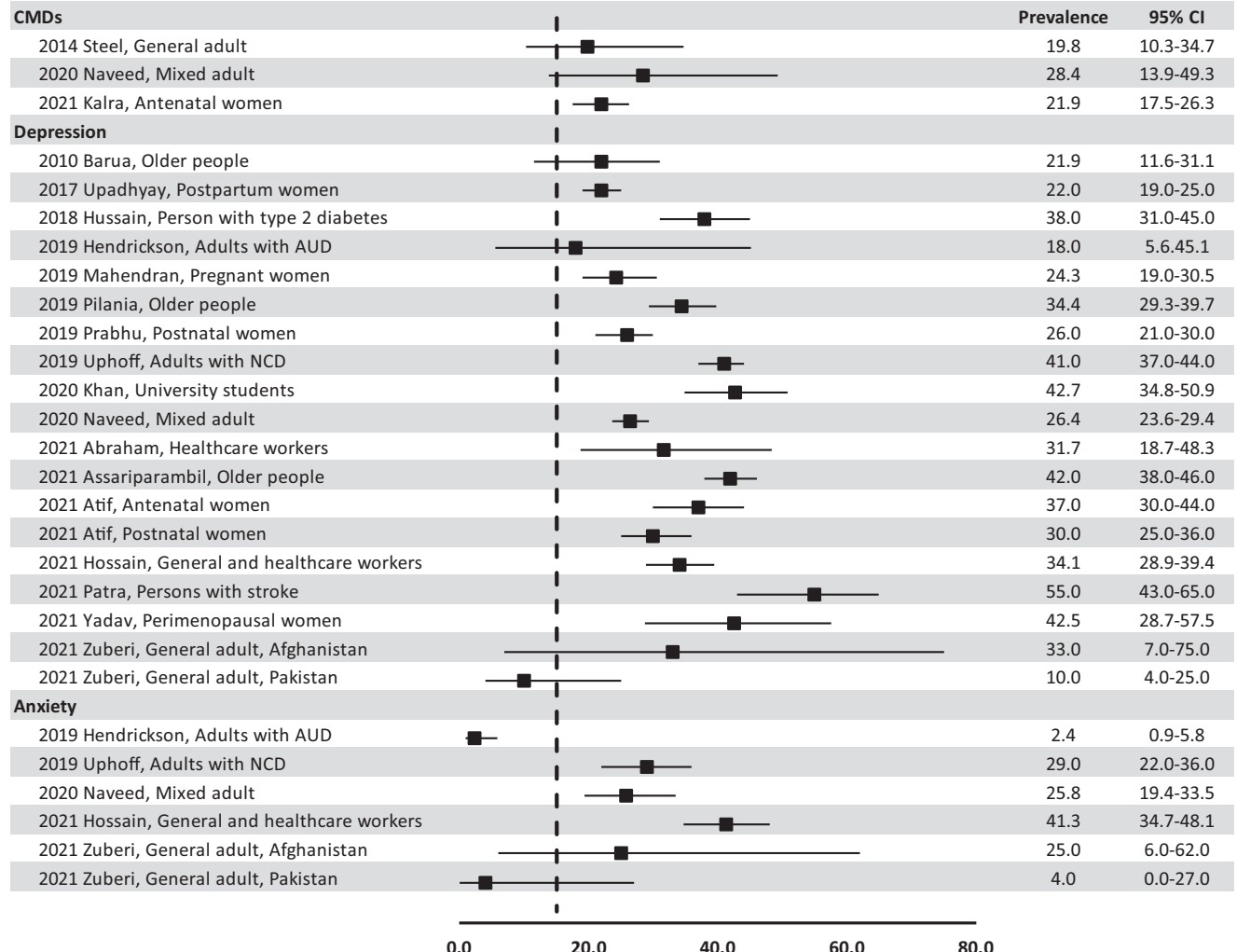

**Figure 2.** Pooled estimates of CMDs, depression and anxiety from meta-analytic reviews.
*Note:* Two studies (2021 Atif and 2021 Zuberi) provided two relevant estimates each for different population groups; the vertical dotted line denotes a pooled prevalence of 14.0% (drawn to correspond with the Figure 3 forest plot).

of databases searched ranged from two to fourteen, and the majority of reviews presented evidence from India (*n* = 90). The number of South Asian primary studies ranged from one to 149, and sample size ranged from 109 to 863,657 (not reported in 15 reviews). Reviews covered diverse populations, with participants recruited from a range of clinical and community-based settings. Only ten

reviews were rated as 'high' quality, while most (61) were rated as 'critically low' (details in Supplementary Appendix 7).

A total of 65 reviews presented the prevalence of mood (affective) disorders including depressive and bipolar disorders, followed by 45 on anxiety disorders, and 10 on a combination of mood and anxiety disorders, grouped together as CMDs. A further nine

reviews reported the prevalence of substance use disorders (SUDs), while others covered a range of other mental disorders: seven on behavioural and emotional disorders with usual onset in childhood and adolescence, including conduct disorder and attention-deficit hyperactivity disorder (ADHD), four on pervasive developmental disorders including autism spectrum disorder (ASD), three each on dementia, schizophrenia and psychotic disorder, personality disorder, and intellectual disabilities, and two on eating disorders. Of these, only one included a meta-analysis providing a pooled estimate of ASD prevalence among children in India. We also found six reviews that reported the prevalence of 'any mental disorder' and 23 that reported on suicide and intentional self-harm. Many identified reviews covered mental disorders in specific population subgroups including older people, perinatal women, students, healthcare workers (HCWs), and persons with comorbidities. Twenty-two reviews focused on the impact of COVID-19 on the psychosocial health of various population groups (Supplementary Appendix 8).

### Summary of pooled prevalence from systematic reviews with meta-analysis

We now focus on the 25 reviews with meta-analyses on the prevalence of various mental disorders in SA. Eleven were exclusively of studies conducted in India, four in Pakistan, and the remaining 10 covered multiple countries in the region. The population comprised all adults (including perinatal women and older people, $n = 1$), general adults ($n = 3$), adults with specific conditions such as alcohol use disorders (AUD) or non-communicable disease (NCD) ($n = 6$), women ($n = 6$), older people ($n = 4$), children and adolescents ($n = 2$), HCWs ($n = 2$), and university students ($n = 1$). In general, these reviews reported high pooled prevalence of mental disorders among both general-adult (up to 33.0% for depression) (Naveed et al., 2020), and specific population subgroups (up to 55.0% for depression among stroke survivors) (Patra et al., 2021). The pooled prevalence of suicidal behaviours among adults was 6.4% (95% CI = 3.1–12.4) (Naveed et al., 2020), and among children and adolescents was 17.1% (5.0–35.4) (Ganesan et al., 2020), whereas the pooled prevalence of any mental disorder among victims of suicide was 90.4% (71.8–97.2) (Cho et al., 2016).

We identified 19 pooled estimates for mood disorders (17 studies), followed by six for anxiety disorders (5 studies), and three for CMDs (Figure 2). The pooled prevalence (range) for depressive disorders in the general population was 10.0% (4.0–25.0) to 33.0% (7.0–75.0). Estimates were generally higher for specific population subgroups, including older people (21.9% (11.6–31.1) to 42.0% (38.0–46.0)), perinatal women (22.0% (19.0–25.0) to 37.0% (30.0–44.0)), peri-menopausal women (42.5% (28.7–57.5)), university students (42.7% (34.8–50.9), HCWs = 31.7% (18.7–48.3) to 34.1% (28.9–39.4)), and adults with comorbidities (18.0% (5.6–45.1) to 55.0% (43.0–65.0)). Similarly, pooled prevalence (range) for anxiety disorders in the general population was 4.0% (0.0–27.0) to 25.8% (19.4–33.5); for adults with comorbidities, it was 2.4% (0.9–5.8) to 29.0% (22.0–36.0), and among all adults and HCWs during the COVID-19 pandemic it was 41.3% (34.7–48.1). Based on reviews covering multiple countries in SA, the pooled prevalence of CMDs in the general adult population alone was estimated to be 19.8% (10.3–34.7) (Steel et al., 2014), whereas it was higher (28.4% (13.9–49.3)) among adult populations that included older people and perinatal women (Naveed et al., 2020). One review from India

reported a pooled value of 21.9% (17.5–26.3) for prevalence of CMDs among antenatal women (Kalra et al., 2021).

We also found one general-adult, population-based estimate for the pooled prevalence of any mental disorder, covering all countries in SA except Maldives and presented as a rate per 1000 (95% CI): 122.0 (8.0–252.0) (Ranjan and Asthana, 2017). In addition, we found two meta-analyses reporting SUDs prevalence of 0.0% (0.0–1.0) to 32.0% (6.0–78.0) (Naveed et al., 2020; Zuberi et al., 2021), one on dementia prevalence (2.0% (2.0–3.0)) (Choudhary et al., 2021) and one on ASD prevalence (0.1% (0.0–0.2)) (Chauhan et al., 2019). These are not presented in Figure 2.

### Pooled prevalence of depression and anxiety in the general adult population from primary studies

We identified 25 primary studies reporting the prevalence of CMDs in the general adult population (Table 2): 16 from India, three from Nepal, one each from Pakistan, Sri Lanka, and Afghanistan, and three large, population-based studies that covered multiple countries in SA. Study quality overall was high. Meta-analyses found a pooled prevalence of 16.0% (95% CI = 11.0–22.0, $I^2 = 99.9\%$) for depression, 12.0% (5.0–21.0, $I^2 = 99.9\%$) for anxiety, and 14.0% (10.0–19.0, $I^2 = 99.9\%$) for depression and anxiety combined (Figure 3).

The pooled prevalence (95% CI) of depression varied notably by country, from 5.0% (4.0–6.0) in Afghanistan, 5.0% (5.0–6.0) in Sri Lanka and 6.0% (5.0–6.0) in Pakistan to 16.0% (10.0–24.0) in India, 25.0% (6.0–52.0) in Nepal, and 25.0% (24.0–25.0) in Bangladesh. Similarly, the pooled prevalence of anxiety varied between 3.0% (2.0–3.0) in Afghanistan, 4.0% (3.0–4.0) in Pakistan and 6.0% (2.0–14.0) in India, to 19.0% (16.0–23.0) in Nepal, 21.0% (20.0–22.0) in Bangladesh, and 65.0% (64.0–66.0) in Sri Lanka. The pooled values for both conditions also varied markedly according to whether (and which) diagnostic or screening tools were used to ascertain the presence of depression and/or anxiety. For depression, the pooled prevalence from estimates based on diagnostic tools (e.g., Composite International Diagnostic Interview (CIDI) and Mini International Neuro-psychiatric Interview (MINI)) was 5.0% (3.0–6.0), whereas it was 27.0% (13.0–44.0) based on screening measures. Similarly, the pooled prevalence for anxiety from estimates based on diagnostic tools was 1.0% (0.0–3.0), whereas it was 26.0% (19.0–34.0) based on screening measures. Funnel plot asymmetry was observed and Egger's test for meta-analysis of depression was statistically significant indicating publication bias. Forest plots for subgroup analyses and funnel plots can be found in Supplementary Appendix 9.

### Discussion

This umbrella review has identified many reviews covering a range of mental disorders in SA, with the majority focusing on the prevalence of CMDs among different population groups. Our findings suggest a high prevalence of these conditions in the region, with greater burden among specific population groups, including perinatal women, older people, people with chronic physical illnesses, refugees, and other vulnerable groups. More than 20 reviews were identified on the prevalence of CMDs during COVID-19 and suggest a high burden of mental disorders among healthcare workers, teachers, and students in SA during the pandemic. In common with Hossain et al. (2020) we found that most studies were from India, while evidence from Afghanistan, Bhutan, and Maldives was particularly limited. The advantages

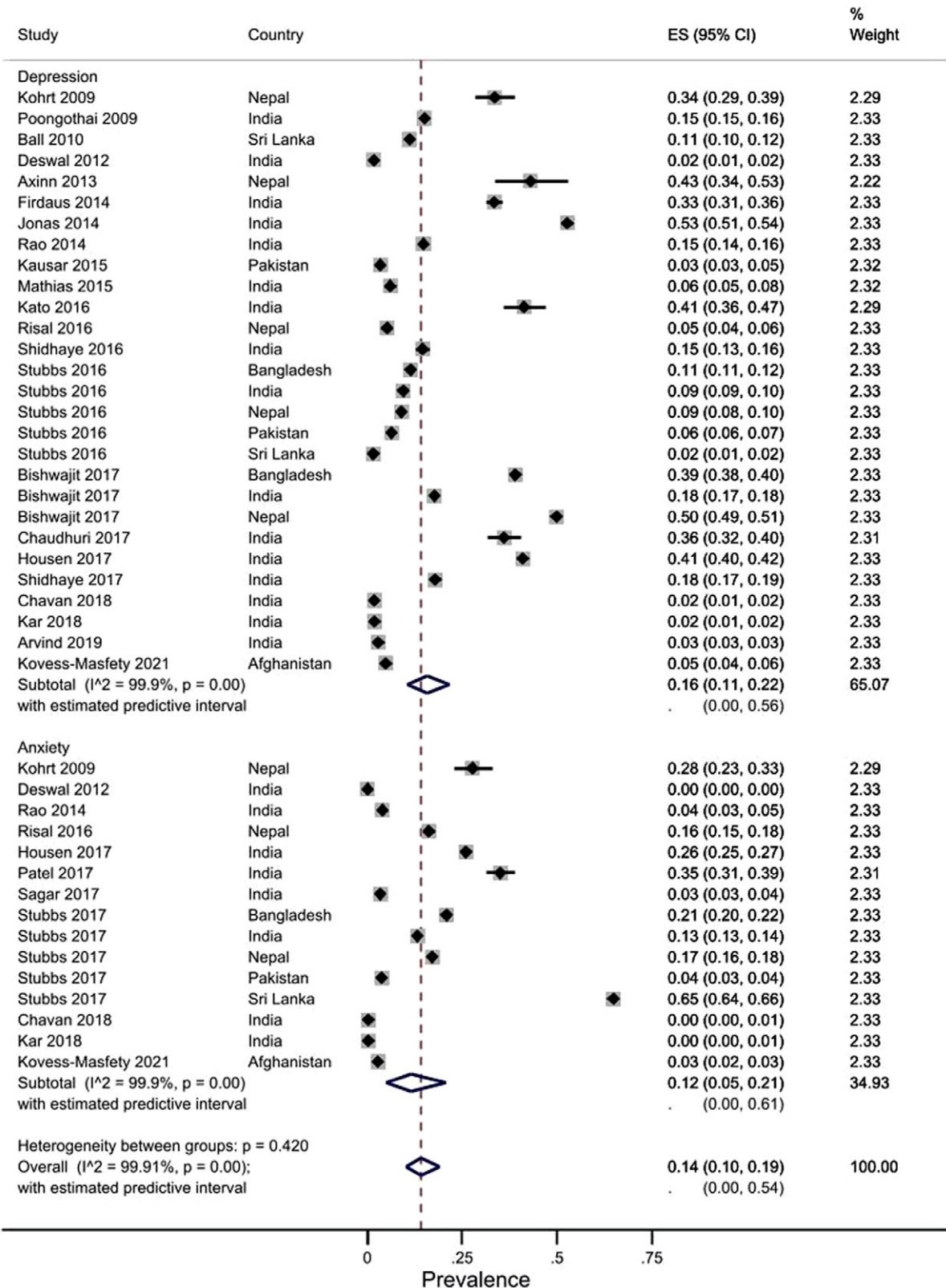

**Figure 3.** Forest plot of primary studies on the prevalence of depression and anxiety in South Asia.

**Table 2.** Summary characteristics of primary studies included in meta-analyses

| Study | Country | Setting | Study design | Sample size | Mental disorder(s) and assessment tools used | Quality score |
|---|---|---|---|---|---|---|
| Kohrt et al. (2009) | Nepal | Mixed | Cross-sectional for prevalence | 307 | Depression – Beck Depression Inventory (BDI); Anxiety – Beck Anxiety Inventory (BAI) | 7 |
| Poongothai et al. (2009) | India | Urban | Cross-sectional | 25,455 | Depression – Modified Patient Health Questionnaire (PHQ12) | 9 |
| Ball et al. (2010) | Sri Lanka | Mixed | Cross-sectional | 5,973 | Depression – Composite International Diagnostic Interview (CIDI) | 9 |
| Deswal and Pawar (2012) | India | Urban | Cross-sectional | 3,023 | Depression, Anxiety – CIDI | 9 |
| Axinn et al. (2013) | Nepal | Rural | Cross-sectional | 400 | Depression – CIDI | 7 |
| Firdaus and Ahmad (2014) | India | Urban | Cross-sectional | 1,326 in 2003; 1,965 in 2013 | Depression – Centre for Epidemiologic Studies Depression Scale (CES-D) | 9 |
| Jonas et al. (2014) | India | Rural | Cross-sectional | 4,711 | Depression – CES-D | 8 |
| Rao et al. (2014) | India | Rural | Cross-sectional | 3,033 | Depression, Anxiety – Mini international neuropsychiatric interview (MINI) | 9 |
| Kausar et al. (2015) | Pakistan | Urban | Cross-sectional | 1,110 | Depression – DSM-based questionnaire | 7 |
| Mathias et al. (2015) | India | Mixed | Cross-sectional | 960 | Depression – PHQ9 | 9 |
| Kato (2016) | India | NR | Cross-sectional | 300 | Depression – PHQ9 and CES-D | 6 |
| Risal et al. (2016) | Nepal | Mixed | Cross-sectional | 2,100 | Depression, Anxiety – Hospital Anxiety and Depression Scale (HADS) | 9 |
| Shidhaye et al. (2016) | India | Rural | Cross-sectional | 1,456 | Depression – PHQ9 | 9 |
| Stubbs et al. (2016) | Bangladesh, India, Nepal, Pakistan, Sri Lanka | Mixed | Cross-sectional | 178,867 | Depression – Based on DSM | 8 |
| Bishwajit et al. (2017) | Bangladesh, India, Nepal | Mixed | Cross-sectional | 14,133 | Depression – Self-reported | 4 |
| Chaudhuri et al. (2017) | India | Mixed | Cross-sectional | 469 | Depression – BDI | 9 |
| Housen et al. (2017) | India | Mixed | Cross-sectional | 5,428 | Depression, Anxiety – Hopkins Symptom Checklist (HSCL-25) | 9 |
| Patel et al. (2017) | India | Urban | Cross-sectional | 605 | Anxiety – State-Trait Anxiety Inventory (STAI) scale | 6 |
| Sagar et al. (2017) | India | Mixed | Cross-sectional | 24,371 | Anxiety – CIDI | 9 |
| Shidhaye et al. (2017) | India | Mixed | Cross-sectional | 3,220 | Depression – PHQ9 | 9 |
| Stubbs et al. (2017) | Bangladesh, India, Nepal, Pakistan, Sri Lanka | Mixed | Cross-sectional | 237,964 | Anxiety – Self-reported | 6 |
| Kar et al. (2018) | India | Mixed | Cross-sectional | 3,508 | Depression, Anxiety – MINI, version 6.0.0 | 9 |
| Chavan et al. (2018) | India | Mixed | Cross-sectional | 2,895 | Depression, Anxiety – MINI, version 6.0.0 | 9 |
| Arvind (2019) | India | Mixed | Cross-sectional | 34,802 | Depression – MINI, version 6.0.0 | 9 |
| Kovess-Masfety et al. (2021) | Afghanistan | Mixed | Cross-sectional | 4,433 | Depression, Anxiety – CIDI | 8 |

and novelty of this review are in providing a more complete and updated picture of the prevalence of mental disorders in the region. But despite the broader inclusion criteria and the updated searches, we found no reviews with pooled estimates of prevalence for many conditions, including severe mental disorders such as schizophrenia and psychotic disorders, behavioural syndromes, personality disorders, or intellectual disabilities. Reviews without meta-analyses for these conditions were also limited. Further, most reviews scored 'low' or 'critically low' on quality assessment,

with very few assessed as providing an accurate and comprehensive summary of available studies on the topic.

Our meta-analysis of primary studies provides pooled estimates for the prevalence of depression and anxiety in the general adult population in SA. We had originally planned to use a 2001 cut-off for the primary studies, set to correspond with the World Health Report on Mental Health (World Health Organization, 2001), but revised this to post-2009 studies, to keep in line with the search period followed by one of the reviews from which we harvested

primary studies (Naveed et al., 2020). Similarly, whilst our protocol mentioned meta-analyses for all mental disorders, we limited this step to reviews on CMDs, given these conditions were the focus of the bulk of identified reviews. Both the reviews from which we harvested primary studies had also previously reported pooled estimates for these conditions in SA, but one included studies in all adult populations, including higher-risk perinatal women and older people (Naveed et al., 2020), while the other was limited to studies in Afghanistan and Pakistan (Zuberi et al., 2021). The inclusion of populations with greater disease burden in the former likely explained its higher prevalence compared to our estimates for both depression (26.4% vs. 16.0%) and anxiety (25.8% vs. 12.0%). With regard to the latter review, while reported country-specific pooled estimates are comparable to ours for Pakistan, its estimates are considerably higher for Afghanistan for both conditions (33.0% vs. 5.0% for depression and 25.0% vs. 3.0% for anxiety). This difference may be explained by the inclusion of two studies reporting high prevalence, which were excluded in our review on the basis of publication date (Scholte et al., 2004; Mufti et al., 2005). On the other hand, our searches identified results from a recent national survey on depression and anxiety disorders in Afghanistan, which we included in our meta-analyses (Kovess-Masfety et al., 2021), while the addition of the excluded primary studies from Afghanistan and Pakistan (Nisar et al., 2004) does not considerably change the region-specific pooled estimates for depression or anxiety (Supplementary Appendix 9).

In addition to mental disorders, our umbrella review included 23 reviews on suicide and intentional self-harm, including one review with meta-analysis among adults in SA, which reported a 6.4% pooled prevalence of suicidal behaviours (Naveed et al., 2020). Other reviews found adult suicide rates ranging from 0.43 to 331.0 per 100,000 population, which varied greatly across countries in the region, and in some cases are likely to be gross underestimations of actual rates (Jordans et al., 2014). An even higher prevalence of suicidal behaviours was found among specific population groups, including perinatal women (Fuhr et al., 2014; Amiri and Behnezhad, 2021), people with HIV/AIDS (Collins et al., 2006; Das and Leibowitz, 2011), female sex workers (Somrongthong et al., 2019) and tribal populations (Devarapalli et al., 2020). Three reviews on suicidal behaviours among children and adolescents were identified, all from India (Aggarwal and Berk, 2015; Aggarwal et al., 2017; Ganesan et al., 2020). Further, we found three reviews among suicide and self-harm populations, which reported a high prevalence of mental disorders, particularly depressive disorders (Cho et al., 2016; Ahmed et al., 2017; Knipe et al., 2019).

Our searches identified three reports based on the Global Burden of Disease studies, which we excluded on the basis of study design (Baxter et al., 2016; Liu et al., 2020; Sagar et al., 2020), and because analyses were either limited to just India or estimated annual percentage change in the burden of depression across the region, not directly comparable to the results of our analyses. Similarly, three reviews (Reddy and Chandrashekar, 1998; Ganguli, 2000; Arora and Aeri, 2019) included in the Hossain et al. (2020) umbrella review did not meet our eligibility criteria on study design, but those topics were covered in other included reviews. Our review includes all other reviews they included, but by going beyond geographically limited reviews and summarising the evidence from multi-country reviews that included at least one South Asian country, we have identified many more reviews, providing a more complete picture of the evidence regarding the prevalence of mental disorders in the region. Diverse terms were used to describe the reviews that were included (systematic, scoping, narrative, etc.), but

we screened for studies that met our criteria to be considered systematic reviews, and thereby ensured consistency in our inclusions (Haddaway et al., 2022). In addition, our meta-analyses of primary studies on depression and anxiety provide important new information on the prevalence of these conditions among the general adult population in the region.

Some key limitations of the research should be acknowledged. First, our approach for identifying primary studies was through harvesting studies from included reviews and forward citation screening, rather than a systematic search and screening of databases. This may have missed studies and introduced a selection bias, but our pre-defined strategy on having a registered protocol likely protected against this. In addition, there are possibilities of publication bias, which our funnel plots suggested were likely. Our meta-analyses also found high heterogeneity, which could be explained to some extent by differences between countries and assessment tool used, demonstrated by subgroup analyses. The finding that studies using screening tools report higher prevalence than those using diagnostic interviews has been previously reported, which may have overestimated the prevalence of mental disorders (Zuberi et al., 2021). In the methodological literature on clinical trials, developing and adopting 'core outcome sets' has been advocated to address the heterogeneity that precludes meaningful synthesis of evidence across studies. Core outcomes sets mandate the inclusion of key outcomes to be measured in all trials of interventions for particular conditions and may also define the tools to be used to measure them (Chiarotto et al., 2017). A similar agreed set of defined measures for observational studies of various mental ill health conditions may be a way forward for better synthesis.

Next, although the majority of primary studies received overall high ratings, few were nationally representative surveys of the general adult population. Nonetheless, there were primary studies from most countries in the region, apart from Bhutan and Maldives. In contrast to the quality of primary studies in our meta-analyses, our narrative synthesis is largely based on reviews that scored 'low' or 'critically low'. We therefore limited our presentation of prevalence estimates solely to the meta-analytical reviews, while the overall narrative summary provides a broader mapping of identified evidence from all reviews by type of review and mental disorder. Finally, there is the possibility that our umbrella review may have missed some relevant reviews on mental disorders in SA, but we searched a large number of (including region-specific) databases and reviewed the literature as comprehensively as possible.

Overall, the findings of our research show a high burden of mental disorders among the general-adult population in SA, with even higher prevalence among specific population subgroups. These findings are also supported by reviews published since our searches were carried out (Manna et al., 2022; Palfreyman and Gazeley, 2022; Al-Mamun et al., 2023; Javan Biparva et al., 2023). Our results highlight an urgent need for countries in SA to formulate and implement both clinical and policy measures for the prevention and early treatment of mental disorders and intentional self-harm. The mapping of evidence according to the type of review and mental disorder (Supplementary Appendix 8) shows that population-level prevalence estimates are generally lacking beyond CMDs, including for schizophrenia and psychotic disorders, behavioural syndromes, personality disorders, and intellectual disabilities. These identified gaps are supported by other recent reviews (Russell et al., 2022; Bastien et al., 2023), and should be a focus of future research, along with the strengthening of epidemiological surveillance systems to better capture morbidity, mortality,

and economic burden of all mental disorders and intentional self-harm in the region.

**Open peer review.** To view the open peer review materials for this article, please visit http://doi.org/10.1017/gmh.2023.72.

**Supplementary material.** The supplementary material for this article can be found at https://doi.org/10.1017/gmh.2023.72.

**Data availability statement.** The details of data searches and extractions from the included studies are provided in the Supplementary Material. The review protocol, including the analysis plan, can be accessed freely from the PROSPERO database, using the registration number mentioned. We do not have any additional data to share.

**Author contribution.** Conceptualization: D.M.D., N.S.; Conducting literature searches: J.W.; Data analysis: A.L.V., M.R.F.; Data extraction and quality appraisal: A.L.V., D.M.D., M.N., K.P.M., S.T., M.R.F., R.H., J.W., S.B., N.S.; Data interpretation: A.L.V., D.M.D., M.R.F., N.S.; Design of literature search strategy: J.W.; Manuscript writing: A.L.V., M.R.F., S.B.; Revision of manuscript and editing: A.L.V., D.M.D., M.N., K.P.M., S.T., M.R.F., R.H., J.W., S.B., N.S.; Study design: D.M.D., N.S. All authors had full access to all the data in the study and had final responsibility for the decision to submit for publication.

**Financial support.** This research is funded by the National Institute for Health Research (NIHR) – Grant 17/63/130 NIHR Global Health Research Group: Improving Outcomes in Mental and Physical Multi-morbidity and Developing Research Capacity (IMPACT) in South Asia at the University of York, using UK aid from the UK Government to support global health research. The views expressed in this publication are those of the author(s) and not necessarily those of the NIHR or the UK government.

**Competing interest.** The authors declare none.

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
