## [Reviewer Report]

To,

The Editorial Team

Global Mental Health

17 April 2023

Subject: A systematic umbrella review and meta-analysis on ‘Prevalence of mental disorders in South Asia’ for your consideration

Dear Editorial Team,

We hereby submit our article titled “Prevalence of mental disorders in South Asia: A systematic review of reviews” to Global Mental Health for your consideration. To our knowledge, this study provides the most comprehensive and up-to-date estimates for the prevalence of mental disorders and intentional self-harm in South Asia.

Through extensive searching and rigorous screening processes, we have identified a large number of systematic reviews reporting the prevalence of mental disorders and/or intentional self-harm in South Asian countries. We have followed the recommended best practices for conducting and reporting our research, and the manuscript is drafted following GMH’s formatting guidelines. In addition to narratively synthesising findings from the reviews themselves, we identified primary studies of common mental disorders among adults from a subset of selected reviews and their forward citations and pooled them using random-effect models. Our findings highlight the need for South Asian countries to formulate and implement both clinical and policy measures for the prevention and early treatment of these conditions within the region.

This research was funded by the UK National Institute for Health Research (NIHR) – Grant 17/63/130 NIHR Global Health Research Group: Improving Outcomes in Mental and Physical Multi-morbidity and Developing Research Capacity (IMPACT) in South Asia at the University of York.

Please let us know if any further information is required.

With many thanks for your consideration,

Yours sincerely,

Aishwarya (on behalf of the review team)

---

## [Reviewer Report]

I appreciate the efforts of the authors to summarize the evidence on the prevalence of mental disorders in South Asia. This systematic review of reviews provides an updated evidence summary that extends both the umbrella review (Hossain et al., 2020) and the meta-analytic review (Naveed et al., 2020) on the same topic. The added value of this review is that offers a methodological integration of these past reviews, through an overview of reviews and also through a meta-analysis of prevalence from empirical works. However, there is a critical concern regarding the conceptualization of this work.

This review of the reviews has adhered to the JBI guidelines (Aromataris, 2015), which clearly stated two fundamental aspects of umbrella review. One, umbrella reviews should include reviews with the same/similar scope- to prevent a mismatch of evidence in context. For example, an umbrella review of an epidemiologic phenomenon in a given population should include reviews that focused on that population only. The inclusion of any review that includes one or a few studies from that population but may also include studies from other geographies or populations can affect the generalizability of evidence if summarized through an umbrella review.

This is critical for umbrella review because in this methodology the “level of synthesis” is “reviews only” – therefore, any review that did not explicitly focus on the population of interest can contaminate the overall evidence presented in the umbrella review.

I appreciate the idea of a meta-analytic review complementing the evidence base of the umbrella review - it is acceptable that the authors may include effect sizes from studies in mixed populations reviews as long as a primary study was conducted in one of the South Asian countries. However, those mixed reviews should not be included in the narrative umbrella review summary or in any pooling approach in the meta-analytic review. Their only possible contribution can be informing the existence of one/more studies/effect sizes within that review. In that case, the respective review should not be included in the last box of the flow chart, rather, it should provide/inform additional article(s) that would enrich the meta-analysis only. This approach, if done carefully, can preserve the value of the meta-analysis as well as provide a clean and intensive overview in the umbrella review.

This is useful from a reader’s perspective as well. A potential reader or user of the review won’t benefit from the summary of the characteristics of “reviews” that may include a significant proportion of studies conducted outside of South Asia. If the authors aim to keep those studies, then the study title should not include South Asia only, which will jeopardize the authors' efforts and the paper’s appeal. Rather, summarizing South Asia-focused papers, plus pooling the prevalence from all primary studies in existing meta-analytic reviews as well as “mixed populations” reviews, can offer an acceptable body of evidence that may inform more accurate insights on mental health in South Asian populations.

I’d recommend reforming the results of the paper and making it consistent with the JBI approach addressing the concerns above. That may reduce the count of reviews in the narrative synthesis, but the value of mixed reviews will be preserved if they can offer studies/effects to the meta-analysis. The inclusion of any reviews with fewer South Asian studies would be enormously problematic, as stated above. The rest of the paper looks excellent; perhaps adding some insights for public mental health research and policymaking- based on the summarized evidence- could strengthen the discussion. Lastly, I congratulate the authors for their contributions to the critical topic, and I hope they’ll revisit their work to make it better from a methodological and practical perspective.

Reference

Aromataris, E., Fernandez, R., Godfrey, C. M., Holly, C., Khalil, H., & Tungpunkom, P. (2015). Summarizing systematic reviews: methodological development, conduct and reporting of an umbrella review approach. JBI Evidence Implementation, 13(3), 132-140.

Hossain, M. M., Purohit, N., Sultana, A., Ma, P., McKyer, E. L. J., & Ahmed, H. U. (2020). Prevalence of mental disorders in South Asia: An umbrella review of systematic reviews and meta-analyses. Asian journal of psychiatry, 51, 102041.

Naveed, S., Waqas, A., Chaudhary, A. M. D., Kumar, S., Abbas, N., Amin, R., … & Saleem, S. (2020). Prevalence of common mental disorders in South Asia: a systematic review and meta-regression analysis. Frontiers in psychiatry, 11, 573150.

---

## [Reviewer Report]

Dear Authors,

This is a competent, thorough and well-rationalised review - and was a pleasure to work through. While I’m conscious some new relevant publications (including systematic reviews for the South Asia region) have come out since your search date in 2021, this is a useful contemporary snapshot of what is available.

Given the classic and persistent challenge of heterogeneity in study tools and design which makes meta-analyses so difficult to do, do signpost to the Appendix table 7 indicating which instrument(s) were used in each primary study. This could then be part of brief, but more explicit reflection on what could be done to move this issue forward around heterogeneity, particularly for under-studied issues (i.e., seemingly all those beyond depression and anxiety).

Stressing the lack of evidence beyond CMDs, the need to move beyond prevalence studies, and heterogeneity of research design would be welcome and Appendix 7 could be informative for those looking to tackle some of these chronic challenges.

Best

---

## [Reviewer Report]

Thank you for submitting your review to Global Mental Health. We have received responses from two reviewers who both noted several strengths of this review, specifically how it builds on prior reviews on this topic. The two reviewers have provided some suggestions regarding the presentation of the results, the methodology, and have requested additional reflections on the measurement considerations. We hope you will consider revising your article based on these suggestions.

---

## [Reviewer Report]

Dear reviewers and editors

Thank you for the opportunity to revise our manuscript as per the constructive editorial and peer-review feedback. We have carefully considered all the feedback received and made appropriate revisions to our manuscript and supplementary material. We trust that these revisions are to your satisfaction. Please do not hesitate to contact us if you require any further information or clarification.

Yours sincerely,

Dr Aishwarya Vidyasaragan (on behalf of all co-authors)

---

## [Reviewer Report]

Dear authors,

The review already displayed competence and thorough thought and attention to detail in round 1. Feedback appears to have been addressed with greater clarity in a number of spots across the manuscript.

There are some 2022/23 systematic reviews inclusive of South Asian country data to my knowledge which would be relevant here (too late of course for inclusion in the analysis itself but could possibly be pulled in for discussion as some takeaways reinforce your recommendations at least for sub-groups, e.g., adolescents and perinatal women). See Gazeley and Palfreyman 2022 for example; there may be forward citations there.

Appendix 8 and the addition of core outcome sets are both constructive; the lean towards talking in terms of transdiagnostic criteria and tools is also relevant here.

Tiny proofreading - check line 293: ‘… we searched a large number OF (including region-specific) databases…’. The word ‘of’ is missing.

I have no additional critique warranting meaningful changes.

Well done.

---

## [Reviewer Report]

The reviewer expressed appreciation for the significant improvements made to the paper based on their feedback. However, they have a couple of minor points for your attention.

---

## [Reviewer Report]

Dear Editors,

We are grateful for the opportunity to revise our paper in light of the reviewers’ helpful feedback. We trust that these revisions are to your satisfaction. Please do not hesitate to contact us if you require any further information or clarification.

Yours sincerely,

Dr Aishwarya Vidyasaragan (on behalf of all co-authors)

---

## [Reviewer Report]

All comments previously shared have been sufficiently addressed. I have no further inputs.

Congratulations to the authors on a thorough and competent review.